# Exploring Mycolactone—The Unique Causative Toxin of Buruli Ulcer: Biosynthetic, Synthetic Pathways, Biomarker for Diagnosis, and Therapeutic Potential

**DOI:** 10.3390/toxins16120528

**Published:** 2024-12-06

**Authors:** Gideon Atinga Akolgo, Kingsley Bampoe Asiedu, Richard Kwamla Amewu

**Affiliations:** 1Department of Chemistry, University of Ghana, Legon-Accra P.O. Box LG56, Ghana; gaakolgo001@st.ug.edu.gh; 2Department of Neglected Tropical Diseases, World Health Organization, 1211 Geneva, Switzerland; asieduk@who.int

**Keywords:** mycolactone, biomarker, Buruli ulcer, toxin, aptamers, immunosuppressive, analgesic and cytotoxic

## Abstract

Mycolactone is a complex macrolide toxin produced by *Mycobacterium ulcerans*, the causative agent of Buruli ulcer. The aim of this paper is to review the chemistry, biosynthetic, and synthetic pathways of mycolactone A/B to help develop an understanding of the mode of action of these polyketides as well as their therapeutic potential. The synthetic work has largely been driven by the desire to afford researchers enough (≥100 mg) of the pure toxins for systematic biological studies toward understanding their very high biological activities. The review focuses on pioneering studies of Kishi which elaborate first-, second-, and third-generation approaches to the synthesis of mycolactones A/B. The three generations focused on the construction of the key intermediates required for the mycolactone synthesis. Synthesis of the first generation involves assignment of the relative and absolute stereochemistry of the mycolactones A and B. This was accomplished by employing a linear series of 17 chemical steps (1.3% overall yield) using the mycolactone core. The second generation significantly improved the first generation in three ways: (1) by optimizing the selection of protecting groups; (2) by removing needless protecting group adjustments; and (3) by enhancing the stereoselectivity and overall synthetic efficiency. Though the synthetic route to the mycolactone core was longer than the first generation, the overall yield was significantly higher (8.8%). The third-generation total synthesis was specifically aimed at an efficient, scalable, stereoselective, and shorter synthesis of mycolactone. The synthesis of the mycolactone core was achieved in 14 linear chemical steps with 19% overall yield. Furthermore, a modular synthetic approach where diverse analogues of mycolactone A/B were synthesized via a cascade of catalytic and/or asymmetric reactions as well as several Pd-catalyzed key steps coupled with hydroboration reactions were reviewed. In addition, the review discusses how mycolactone is employed in the diagnosis of Buruli ulcer with emphasis on detection methods of mass spectrometry, immunological assays, RNA aptamer techniques, and fluorescent-thin layer chromatography (f-TLC) methods as diagnostic tools. We examined studies of the structure–activity relationship (SAR) of various analogues of mycolactone. The paper highlights the multiple biological consequences associated with mycolactone such as skin ulceration, host immunomodulation, and analgesia. These effects are attributed to various proposed mechanisms of actions including Wiskott–Aldrich Syndrome protein (WASP)/neural Wiskott–Aldrich Syndrome protein (N-WASP) inhibition, Sec61 translocon inhibition, angiotensin II type 2 receptor (AT2R) inhibition, and inhibition of mTOR. The possible application of novel mycolactone analogues produced based on SAR investigations as therapeutic agents for the treatment of inflammatory disorders and inflammatory pain are discussed. Additionally, their therapeutic potential as anti-viral and anti-cancer agents have also been addressed.

## 1. Introduction

Buruli ulcer (BU) is a destructive mycobacterial disease of the subcutaneous tissue caused by *Mycobacterium ulcerans* (MU)*. M. ulcerans* is related to the family of acid-fast bacilli mycobacterial diseases such as *Mycobacterium tuberculosis* and *Mycobacterium leprae* that cause tuberculosis and leprosy, respectively [1,2]. In contrast to tuberculosis (TB) and leprosy, BU is the least common disease worldwide, and human-to-human transmission of BU is uncommon [3]. BU has been reported to be endemic in at least 33 tropical and subtropical countries globally with the highest prevalence and incidence occurring in West and Central African countries [4,5]. Unlike Australia where the disease is more common in the adult population, it primarily affects children in Africa, especially those aged 5 and 15 [6].

BU typically starts as a small painless subcutaneous papule or nodule, or other pre-ulcerative forms like plaques, and oedema that progress into large necrotic skin ulcerations with distinctly undermined edges that may cover 15% of a patient’s skin, if treatment is not given (Figure 1) [7,8,9,10].

Extreme forms of disease such as osteomyelitis extend to the bones of infected persons. Large and severe ulcers result in protracted healing which leads to significant cosmetic and functional deformities and disabilities, particularly in children, who if healed, leave behind scarring, disfigurement, and joint contractures [4]. As a result of the devastating consequences presented by the disease, the World Health Organization (WHO) has listed BU as a top-ranking and emerging neglected tropical disease (NTD) [11,12,13].

Ghana, the Democratic Republic of the Congo, Benin, and Côte d’Ivoire are among the West and Central African nations where the disease is most endemic [4]. The disease is more severe among those living in poverty and more common in rural parts of Africa [14,15]. In endemic hospitals in Africa, the disease accounts for approximately 30% of ulcer cases, even though it is significantly under-reported [16]. In 2023, 1952 new cases were reported from 12 countries, with 1573 of the reported cases from the African Region alone and 379 from the Western Pacific Region [17] (Figure 2).

## 2. Etymology of Buruli Ulcer

Historically, there is an intriguing connection between Buruli ulcer and Nile mythology. Early in the 1860s, the Royal Geographical Society of London organized an expedition and in October of the same year, Captains John Hanning Speke and James Augustus Grant set out from Zanzibar “with the view to discover, if possible, the sources of the Nile” [18]. Their expedition confirmed that the White Nile does indeed originate in Lake Victoria (previously Lake Nyanza) and more precisely the Ripon Falls at the Lake’s northern end. Captain James Augustus Grant wrote about their expedition in search of the White Nile’s source. Sadly, Grant contracted a mystery disease in December 1861 and was bedridden for five months, so he was not on the shores in July 1862 when Lake Victoria was named as the source of the Nile. From Grant’s intriguing narrative of these explorations which was published in 1864 titled, *A Walk Across Africa or Domestic Scenes from My Nile Journal*, he described how the infection that he contracted in December 1861 kept him from joining Speke on the shore of Lake Victoria in July 1862 [19]. In the detailed first-person narrative of his ailment, he wrote: “The right leg, from above the knee, became deformed with inflammation, and remained for a month in this unaccountable state, giving intense pain, which was relieved temporarily by a deep incision and copious discharge. For three months, fresh abscesses formed, and other incisions were made; my strength was prostrated; the knee stiff and alarmingly bent, and walking was impracticable” [20]. Captain Grant might have described what is regarded today as the first account of the earliest reported case of Buruli ulcer. This vivid description showed similar indications of the oedematous form of Buruli ulcer that occurs in Western and Central Africa [21].

In 1897, a British missionary physician Sir Ruskin Albert discovered Buruli ulcer at Mengo Hospital in Kampala, Uganda and provided the first clinical description of the disease. In his case report, Cook detailed uncommon ulcers from Kampala Hospital in Uganda with admission diagnosis of “tubercular ulceration of arms and legs” [22,23,24]. Later, from 1920–1935, Kleinschmidt, another missionary physician in northeast Zaire (now Democratic Republic of the Congo), also indicated the presence of the disease in Uganda and northeast Zaire. Kleinschmidt observed ulcers with large numbers of acid-fast bacilli and undermining skin [25].

As early as 1937, the disease was identified in Bairnsdale in Victoria, southeast Australia by two medical physicians, Alsop and Searls. An epidermic that was later confirmed as BU disease broke out in 1939 following catastrophic rainfalls in the state of Victoria in 1935 [26]. However, it was only in 1948 that Australian pathologist Peter MacCallum and colleagues who were caring for patients at Bairnsdale Hospital provided the first thorough description of BU disease. In a seminal report, they described in detail the histopathological characteristics of six Australian patients from rural riverine regions who each had an unidentified single ulcerative lesion with undermined edges on an arm or a leg. The histopathologic findings included extensive necrosis and a large number of acid-fast bacilli without the formation of granulomas [1,27]. Although there was no evidence of tuberculosis, the ulcers had undermined edges, but with no other signs of tuberculosis except one patient who had a positive tuberculin skin test result. Acid-fast bacilli were present in histological specimens [28]. Initial attempts to cultivate the bacterium with different types of media, atmospheres, and temperatures failed but were finally achieved serendipitously with the inadvertent use of a faulty incubator delivering a controlled 33 °C [24]. It was therefore realized that in contrast to *M. tuberculosis* which can be grown at 37 °C, *M. ulcerans* requires optimal temperatures above 25 °C, but below 37 °C (ideally 30–33 °C) for growth although this growth is quite slow with a doubling time between thirty-six and eighty hours [1,29]. This optimum temperature is consistent with the primary localization of the microorganisms in the subcutaneous tissue and lower dermis of infected humans, as suggested by Buckle and Tolhurst in the original 1948 report [29] and later confirmed by Fenner in 1956 [30]. Additionally, it was later demonstrated that low oxygen levels were crucial for cultivating this extremely slow-growing mycobacterium [31].

Since the establishment of the aetiology of the disease by MacCallum and co-workers among a small group of six patients in the Bairnsdale region of Victoria, Australia, the mycobacterium was tentatively dubbed “Bairnsdale ulcer”, after the region where five of the six patients resided [1]. A few years after they were first described, *M. ulcerans* cases were also observed in Uganda [32,33,34] and the modern-day Democratic Republic of Congo [35]. The name Buruli ulcer was proposed after the Buruli County in Mengo district (today Nakasongola district) near Lake Kyoga in Uganda, a semi-arid area where large numbers of cases were reported in the 1960s [36,37]. The term “Buruli ulcer” was thus first proposed in 1972 by Clancey, Dodge, and Lunn in Uganda following outbreaks in other African countries [32,36]. Searls ulcer or Daintree ulcer are other names of Buruli ulcer. Chronologically, despite WHO approving the name Buruli ulcer, Bairnsdale ulcer would also have been historically a correct denomination [38].

Buruli ulcer have a history in Africa that can be traced to pre-1980 and after 1980. Before 1980, several African nations, including Cameroon, the Democratic Republic of Congo, Gabon, Ghana, Nigeria, and Uganda, published significant works on the occurrence of the disease. Cases were suspected but never verified in the United Republic of Tanzania, Kenya, Sudan, and the Central African Republic [4]. The most significant prevalence came from the Democratic Republic of the Congo and Uganda [36]. West Africa saw the emergence of new Buruli ulcer foci after 1980. In many West African nations, particularly Benin, Côte d’Ivoire, Ghana, and more recently, Angola, Burkina Faso, Guinea, and Togo, there has been a sharp rise in the incidence of disease [4,39,40].

## 3. *Mycobacterium ulcerans* and Its Unique Toxin Mycolactone

### 3.1. Mycobacterium ulcerans

*Mycobacterium ulcerans,* first identified in 1947, is a pathogenic non-tuberculous mycobacterium (NTM) that causes Buruli ulcer [41]. *M. ulcerans* is a bacterial pathogen that is acquired from the natural environment. It has long been associated with aquatic and swampy environments, especially those that are close to slow-moving watercourses [42]. *M. ulcerans* is known to adapt to certain ecological niches and is believed to reside in natural environments in which pH, oxygen, and nutrient availability are likely to regulate its growth and mycolactone production [31,43,44,45]. Additionally, evidence suggests that living species including fishes and amphibians [46], mosquitoes [47], and aquatic insects [48] can host and spread it. Although *M. ulcerans* has long been linked to wetlands and riverine regions as natural reservoirs [34,49], the disease is sometimes referred to as the “mysterious disease” due to the fact that its mode of transmission is still elusive and remains obscure despite several hypotheses being proposed [50,51,52].

### 3.2. Mycolactone: The Unique Toxin Behind Mycobacterium ulcerans

Even though the majority of pathogenic bacteria produce toxins that are crucial to their pathogenicity, there is currently no proof that the two other most well-known pathogenic species of the genus Mycobacterium—*Mycobacterium tuberculosis* and *Mycobacterium leprae*—produce toxins. Consequently, *Mycobacterium ulcerans* is distinct from other human infections in that it is the only species of the mycobacterium family that produces and secretes a potent cytotoxin [53]. In 1965, Connor and colleagues suggested that *Mycobacterium ulcerans* played a key role in producing the exotoxin [54]. Later, in 1974, two follow-up reports by Conner and colleagues supported this hypothesis by showing that injecting culture filtrates of different *M. ulcerans* strains into footpads of mice and guinea pig epidermis produced outcomes that were similar to those generated by inoculation of animals with the actual organism [55,56]. However, the actual nature of the toxin was only established in 1998 by Small and co-workers [57]. The toxin was effectively isolated, purified from the acetone-soluble lipid extracts of *M. ulcerans* and characterized initially by thin-layer chromatography (TLC) and mass spectrometry (MS). The toxin was detected by TLC as a light-yellow band with a retention factor (*R*_F_) of 0.23 using chloroform: methanol: water (90:10:1) as the solvent system [57]. MS analysis of the toxin under electrospray ionization in positive ion mode showed an abundant sodium adduct [M + Na]^+^ of mycolactone signal at *m*/*z* 765.5. Subsequently, the complete chemical structure of the toxin was solved to be a 12-membered ring polyketide macrolide called mycolactone (Figure 3) [57,58,59].

Mycolactone is crucial for the virulence and pathogenesis of *M. ulcerans* disease and hence appropriately classified as the toxin that causes Buruli ulcer [10,60]. The toxin has been reported to be responsible for tissue necrosis, painlessness, and immunosuppressive properties [57,61,62,63,64]. This has been demonstrated in animal models, where studies have revealed that guinea pigs given an intradermal injection of pure mycolactone developed lesions that closely resembled those seen in human Buruli ulcer lesions [10]. Mycolactone is linked to vacuolar nerve tissue damage in mice, which may explain why BU lesions are painless [65,66]. Additionally, mycolactone has also demonstrated cytotoxic and immunosuppressive characteristics and has thus shown to cause necrosis and ulceration [64,67,68]. Many mammalian cell types such as macrophages [69,70,71,72,73], myocytes [74,75], fibroblasts [10,76,77,78,79], keratinocytes [67], dendritic cells [80], T-cells [81,82], and adipocytes [83] have also been demonstrated to be affected by mycolactone both in vitro and in vivo. The effect of mycolactone on these cells include decreased cytokine production and interference with cellular signaling [84], cytoskeletal reorganizations, necrosis, or induction of apoptosis in mammalian cells [85], and lastly, a downregulation of both local and systemic immune responses by interfering with immune cell activation [76,86].

The species of pathogenic mycobacteria that are responsible for the production and secretion of mycolactones are very closely related in their evolutionary history. Together, these mycobacteria are known as mycolactone-producing mycobacteria (MPM). They have been given several other names such as *M. ulcerans*, *M. pseudoshottsii*, *M. liflandii*, and some strains of *M. marinum* [87,88,89]. Structurally, ten distinct variants of mycolactones encompassing nine naturally occurring members and one genetically engineered *Marinum* strain have been described so far. Remarkably, all known mycolactones amongst different *M. ulcerans* strains appear to contain the same macrolide core composed of an invariant core comprising a 12-membered lactone ring (Figure 3, labeled red) with a C11-linked northern acyl side chain (Figure 3, labeled blue) but each is attached to a different highly polyunsaturated southern acyl side chain that is esterified via a C5–O linkage (Figure 3, labeled black) [61]. Therefore, while the macrolactone core structure together with the upper side chain are completely conserved, mycolactone populations from various *M. ulcerans* sub-lineages differ in terms of the length, the number of hydroxyl groups and their locations, as well as the number of double bonds in the bottom side chain [90]. Subsequent research has shown that distinct alterations in the biosynthesis process may be responsible for the variety of the different toxins from diverse geographic sources with differences inside chain configurations [47,48].

**Figure 3 toxins-16-00528-f003:**
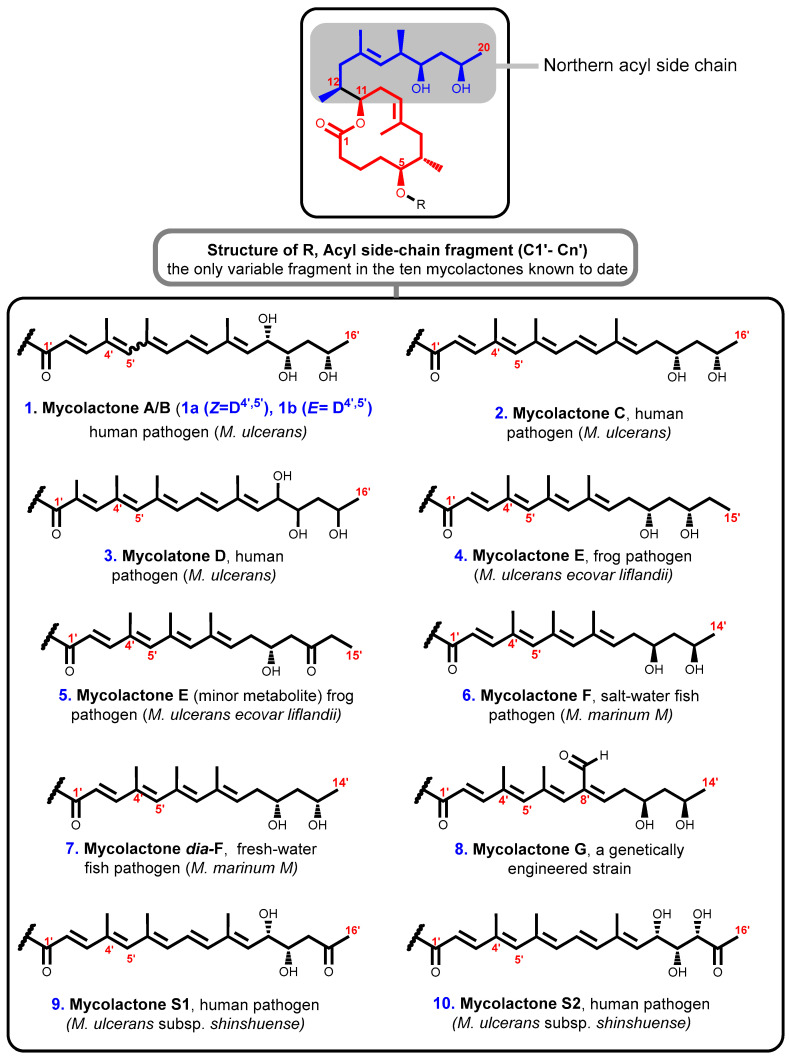
Structures of mycolactones A/B [89], C [90], D [91], E [91,92,93], E [91,92,93], F [78,94], *dia*-F [95,96], G [97], S1 [98], and S2 [98]. Lactone ring highlighted in red, C-linked C12–C20 side chain highlighted in blue, and polyunsaturated fatty acid side chain is in black.

All mycolactones have been shown to have cytotoxic [61], immunosuppressive [61,64], and apoptosis-inducing [10,67] effects. However, it has been noted that the potency levels of various individual structures vary noticeably. The most potent toxin, for instance, is believed to be a stereoisomeric combination of mycolactones A and B that was isolated from the classical *M. ulcerans* lineage from West Africa; the potency of the other mycolactones decreases as they are listed alphabetically [97]. Also, the extremely virulent strains of *M. ulcerans* seen in Africa and Malaysia produce the largest quantities of mycolactone A and B [90].

### 3.3. Gross Structure of Mycolactones A and B

The gross structure of mycolactones A and B is composed of three different sectors: a 12-membered macrocyclic lactone core C1–C11 with two laterally attached side chains; a short northern C-linked side chain fragment comprising C12–C20; and a much longer southern C5-O-linked polyunsaturated, or fatty acid C1′–C16′ acyl side chain fragment. The C1′–C16′ acyl side chain is a sensitive conjugated pentaenoic acid ester possessing three stereogenic centers at C12′, C13′, and C15′ [84,99]. The stereochemistry, relative and absolute configurations, and the complete structure of the mycolactone core have been established and confirmed by Nuclear Magnetic Resonance (NMR) database and a newly developed universal database concept in chiral solvents, respectively, by Kishi’s group [100,101,102]. Subsequently, the assigned structure was confirmed by the preparation of model compounds through total synthesis [103,104]. Importantly, works from these have led to the characterization of mycolactone A and B under standard laboratory conditions, as a mixture of two stereoisomers in an approximately 3:2 rapidly equilibrating mixture of *Z*-Δ^4′,5′^ (mycolactone A) and *E*-Δ^4′,5′^ (mycolactone B) geometric isomers centered around the double bond at C4′ C5′ of the unsaturated side chain having the stereochemistry as shown (Figure 4) [101,102].

### 3.4. Biosynthesis of Mycolactone

*M. ulcerans* is the only species of the mycobacteria family known to produce mycolactone, a unique macrolide toxin that causes the subcutaneous adipose tissue around bacterial colonization sites to necrotize [57,83,105]. *M. ulcerans* and closely related mycobacteria produce polyketide macrolide through the production of polyketide synthases (PKS), which are encoded on the giant 174 kb extrachromosomal plasmid known as pMUM001 [106,107]. There are three enormous genes (mlsA1: 51 kb, mlsA2: 7.2 kb, mlsB: 42 kb) present in mycolactone PKS modules and their constituent domains that encode for three Type I PKS of striking sizes, 1.8 MDa, 0.26 MDa, and 1.2 MDa for MLSA1, MLSA2, and MLSB, respectively [108]. The enzymes ketosynthase KS, acyltransferase AT, keto-reductase KR, dehydratase DH, and enoyl reductase ER present in the module of domains of MLSA and MLSB allow for the synthesis and functionalization of the polyketidic chains (Figure 5a). The PKS has a number of extremely peculiar characteristics that significantly impact the structural underpinnings of the polyketide chain growth specificity on these multienzymes [109,110,111]. First, two of the multi-locus sequence PKS proteins; MLSA1 and MLSA2, are composed of loading module (one), extension modules (nine), and a terminal thioesterase (TE), whilst MLSB consists of one each of loading module and thioesterase and seven extension modules. More surprising still is the extreme mutual sequence similarity between comparable domains in all 16 chain-extension modules suggestive of very recent evolution [108]. The novel SigA-like promoter sequence is required for the transcription of PKS [108,112].

The proposed initial steps in the generation of the macrolide toxin involve the process where *mlsA1* and *mlsA2* synthesize the upper side chain and mycolactone core (C1–C20 fragment) of the mycolactone [109,110,111] whilst *mlsB* synthesizes its C1′–C16′ acyl moiety [53,111]. A putative *beta*-ketoacyl transferase encoded by pMUM gene, mup045, catalyzes the ester linkage between the acyl side chain and the macrolactone core whilst a P450 hydroxylase, encoded by mup053, is responsible for the oxidation of the side chain at C12′ (Figure 5b) [88,91,107]. A third gene, mup038, suggested to encode a type II thioesterase is required for eliminating the aberrant polyketide extension products from the *mls* PKS that form during synthesis. A characteristics feature of the mycolactone PKS is the unexpected high level of sequence identity between domains of the same function (98.7–100% and 98.3–100% for nuecleotide) and amino acid identities, respectively) [53]. This observation suggests a recent evolution of the locus and may be prone to rearrangements that will lead to either loss of in the level of mycolactone or production of new mycolactones. These hypotheses have gained support as recent studies have shown that (i) all mycolactone-producing mycobacteria comprising *M. ulcerans* and those closely related fish and frog pathogens evolved from a common *Mycobacterium marinum* ancestor by pMUM plasmid acquisition [88,113], (ii) formation of mycolactone negative mutants via spontaneous deletion of *mls* gene fragments from laboratory passaging [106], and (iii) swapping of acyltransferase and ketoreductase domains naturally has afforded new mycolactones due to a loss or gain of the entire extension modules in some strains of *M. ulcerans* [89,91].

**Figure 5 toxins-16-00528-f005:**
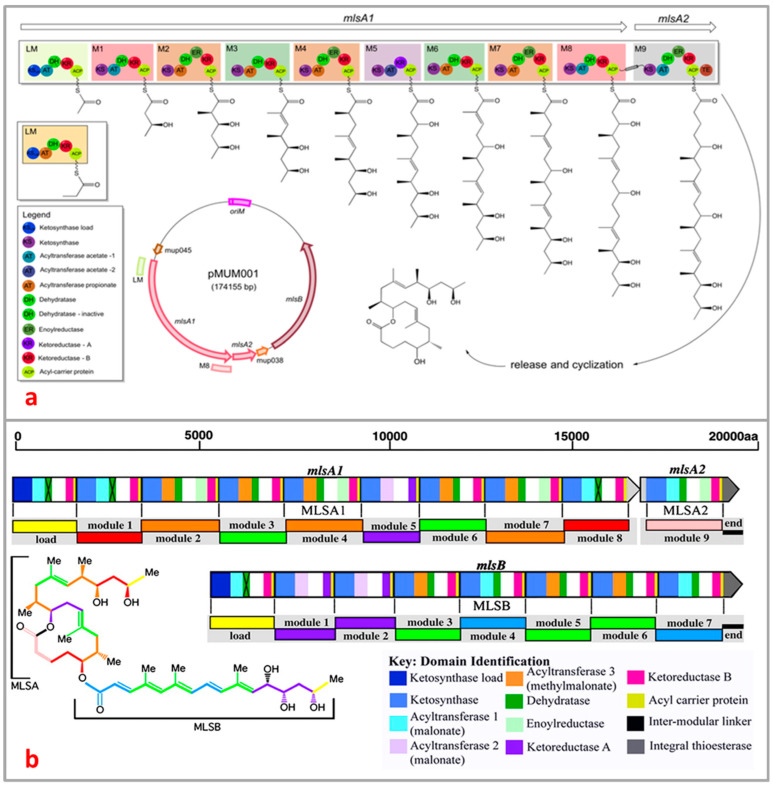
Overview of domain and module organization of the mycolactone PKS genes (**a**) *MlsA1* and *MlsA2* from the mycolactone PKS, harbored by the plasmid pMUM001 from *M. ulcerans* Agy99 [111,114]; (**b**) subunits (MLSA1, MLSA2, and MLSB) of different domains are represented by color block [115].

### 3.5. Chemical Synthesis of Mycolactone A and B

Several biological studies have provided an early understanding of the role mycolactone plays in Buruli ulcer [66,69,70,72,76,80,116,117,118,119]. Studies have shown that increasingly, larger quantities (≥100 mg) of mycolactone A/B are required to help develop an understanding of the mode of action of these polyketides. Unfortunately, investigations of the various mycolactone structures and the explorations of their biological potencies have been seriously hampered by the slow growth of the host mycobacterium and the small (usually microgram) quantities of the toxin available from laboratory-scale culturing efforts, respectively [120]. This has further been complicated by the fact that analysis of culture extracts of a typical strain of *M. ulcerans* revealed the presence of minor amounts of additional mycolactones, differing from mycolactone A and B only in the side chain thus, resulting in a clear difference between the in vivo and in vitro activity of mycolactone A/B [10,61,66,75,90]. For these reasons, interest from the scientific community on mycolactone is at a record high [121,122,123]. Even though the total synthesis of mycolactone is an extremely complicated process requiring substantial synthetic expertise, and thus limiting the production of this crucial toxin for research purposes [38,99], the Kishi group made an outstanding contribution to mycolactone chemistry, synthesis, and research [63,102,103,104]. Considering that mycolactone A/B were the first examples of polyketide macrolides to be isolated from a human pathogen, the synthetic work has largely been driven by the desire to afford researchers enough pure toxins as well as their analogues for systematic biological studies for their highly potent biological activity [121,122,123]. The first total synthesis of mycolactone A and B was achieved by the Kishi group, who also established the relative and absolute configurations and stereostructure of these natural products by chemical synthesis and extensive NMR spectroscopic studies [63,102,103,104]. The Kishi group has elaborated first- [102,104], second- [103], and third-generation [124] approaches to the synthesis of mycolactone A/B as well as the syntheses for mycolactone C [125], E [126], and F [96]. These ground-breaking works have offered efficient synthetic mechanisms of mycolactone A/B [63,103]. Following these pioneering research, several research groups; Negishi [127], Altman [128], Burkart [123,129], Gurjar [130], Feringa/Minnaard [121], Blanchard [131], Dai [132], and Aggarwal [133] have reported total and partial syntheses of the various mycolactones. More recently, modular synthetic approaches for the rapid production of diverse analogues of mycolactone A/B have also been described by Saint-Auret and co-workers [99,131,134,135]. Herein, we discuss in detail the generations of total synthesis of mycolactone by the Kishi group.

#### 3.5.1. First-Generation Total Synthesis (2001–2002)

In 2002, Kishi and co-workers reported the first synthesis of mycolactone A and B [102]. The mycolactone core 11 (Figure 6) had previously been synthesized by Benowitz et al. (2001) for purposes of stereochemical assignment [102]. The synthetic approach employed provided a very adaptable way to synthesize each diastereomer by selecting the right stereochemistry for each component. The synthesis of the core involved the application of Pd-catalyzed isobutylalkenyl Negishi-typed cross-coupling [136,137] at two critical stages, which was to be cyclized by macrolactonization [102]. However, only one disconnection has been adopted for the first-generation Kishi construction of the five C=C bonds of the C1′–C16′ pentaenoic acid fragment, a Horner–Wadsworth–Emmons (HWE) carbonyl olefination reaction for the C8′–C9′ bond (Figure 6) [138,139].

#### 3.5.2. First-Generation Synthesis of the Mycolactone Core

The first-generation synthesis of the mycolactone core **11** was accomplished through the syntheses of various intermediates C1–C7 (Figure 1), C8–C13 (Figure 2), and C14–C20 (Figure 3). Upon completion, the intermediates were coupled by employing modification of Negishi’s conditions to give mycolactone core **11** (Figure 4) [140,141,142].

Synthesis of the C1–C7 fragment

The synthesis of the C1–C7 alkyl iodide **14** was accomplished using *tert*-butyldimethylsilyl (TBS)-protected 5-hydroxypentanal **13** as starting material which progressed via an asymmetric Brown crotylation leading to the formation of the C5 and C6 stereocenters (Figure 1).

Synthesis of the C8–C13 fragment

Using TBDPS-protected (*R*)-hydroxy-2-methylbut-3-ene **15** produced in accordance with literature protocols [142], vinyl iodide **16**, which is made up of the C8–C13 segment was synthesized, establishing the stereochemistry at C12. The vinyl iodide moiety was produced by a (poorly diastereoselective) epoxidation, epoxide opening with a propynyl anion, and a hydrozirconation/iodination reaction as part of the six-step process from **15** to vinyl iodide **16** (Figure 2).

Synthesis of the C14–C20 fragment

Vinyl iodide **20**, corresponding to the C14–C20 portion of the core extension, was synthesized from literature-known (*R*)-3-((tert-butyldimethylsilyl)oxy)butanal, (*R*)-**17**, which defined the configuration of the C19 stereocenter. Aldehyde (*R*)-**17** was subjected to an asymmetric Brown crotylation reaction [143,144] to establish the C16 and C17 stereocenters (Figure 3).

#### 3.5.3. Assembly of the Mycolactone Core

The first-generation synthesis of the mycolactone core **11** was accomplished by coupling the C1–C7 alkyl iodide **14** (Figure 1) with the C8–C13 vinyl iodide **16** (Figure 2) employing a modification of Negishi’s conditions to give **21** [137,140] (Figure 4). This compound was then subjected to five-step sequence reactions to generate iodide **22**, which was then coupled with the C14–C20 vinyl iodide fragment **20** (Figure 3) via a second modified Negishi coupling to give **23** [136,137,145] (Figure 4). Both the triethylsilyl (TES) ether and the primary tert-butyldimethylsilyl (TBS) ether were then selectively removed with buffered HF•pyridine. The primary alcohol was selectively oxidized first to the aldehyde with 2,2,6,6-tetramethylpiperidine 1-oxyl (TEMPO), and then to the carboxylic acid with buffered NaClO_2_ to give **24**. Yamaguchi macrolactonization proceeded smoothly to give the lactone, and a two-step deprotection protocol afforded the C1-C20 diastereomer **11**. Selective protection of the two hydroxyl groups in 1,3-diol positions at C17 and C19 of **11** was accomplished by treatment with dimethoxycyclopentane **25** and *p*-TsOH following the protocol used by Kishi, to afford the titled suitably protected core **26** in good yield [103,104].

#### 3.5.4. Second-Generation Total Synthesis

In 2007, Kishi and co-workers reported an improved and shorter synthesis of mycolactone based on the knowledge accumulated in their previous work [104]. This was achieved by replacing the more challenging acid-promoted deprotection of the cyclopentylidene following the discovery that mycolactone A and B is stable to tetra-*n*-butylammonium fluoride (TBAF)-promoted TBS-deprotection conditions. Thus the cyclopentylidene protecting group present in the C17/C19-diol **26** (Figure 4) was replaced by TBS ethers [103] as seen in **35** (Figure 5), which was subjected to deprotection using tetra-*n*-butylammonium fluoride (TBAF).

#### 3.5.5. Third-Generation Total Synthesis

Aiming to shorten and improve upon the production of mycolactone A/B, further attempts were made by Kishi and co-workers through a scalable construction of the core macrocycle **43** fragments (with C12–C13 already attached) by Yamaguchi macrolactonization, the establishment of the complete C12–C20 side chain through the formation of the C13–C14 bond by Negishi coupling, and attachment of the C1′–C16′ acyl side chain using Yamaguchi esterification as a final step [124]. The synthesis of the acyl fragment relied heavily on Horner–Wadsworth–Emmons (HWE) chemistry for double-bond construction. The first-generation synthesis allowed for the unambiguous confirmation of the relative and absolute stereochemistry of the mycolactone. The second generation focused on the use of optimized protecting groups for the selective protection of the two hydroxyl groups in the 1,3-diol positions at C17 and C19 on the northern acyl side chain of the mycolactone core as TBS-silyl ether [102,103,104]. The synthetic efforts of the third generation allow a scalable and efficient route to the mycolactone core which led to the preparation of multi-gram quantities in high purity [124].

The requisite fragments **36** (C1–C7), **37** (C8–C13), and **44** (C14–C20) were separately synthesized and subsequently assembled to construct the mycolactone core **45** (Figure 6). The coupling of the first two fragments, **36** and **37** to form **38**, was accomplished using a Negishi coupling (Figure 6) [136]. Fortunately, the cyclopentylidene acetal was cleaved under acidic conditions to form **39**, which was subsequently converted to *seco*-acid **40** in a two-step sequence consisting of primary alcohol protection and saponification of the methyl ester. The selective protection of the primary alcohol was cleanly achieved using triisopropylsilyl triflate and 2,6-lutidine at −78 °C. On exposure to standard Yamaguchi conditions, *seco*-acid **40** was converted to macrolactone **41**. The synthesis of the mycolactone core was completed when the triisopropyl ether on **41** was cleaved to form a primary alcohol **42**. The resultant primary alcohol, **42**, was converted to alkyl iodide **43**. Under optimized conditions of Negishi coupling, alkyl iodide **43** was coupled with **44** to provide mycolactone core **45** in 88% yield (Figure 6).

#### 3.5.6. Synthesis of a Suitably Protected Pentaenoic Acid

As stated earlier, the mycolactone family encompasses nine members of natural occurrence (A/B, C, D, E, E-minor metabolite, F, *dia*-F, S1, and S2) and one genetically engineered *Marinum* strain (mycolactone G) that differ only in the nature of the polyunsaturated southern acyl side chain fragment, while the macrolactonic and northern fragments are conserved (Figure 3). This striking structural heterogeneity in the length and substitution pattern results in different biological activities. Hence, a synthetic approach of the southern fragment is required for the rapid production of diverse analogues that would be structurally relevant to the exploration of the biology of *M. ulcerans* infections.

It was anticipated that the pentaenoate system **12** would be unstable hence the focus was to synthesize a suitably protected pentaenoic acid. Application of Horner–Wadsworth–Emmons olefination at C8′–C9′ (Figure 7) afforded the synthesis ethyl ester of the pentaene fatty acid as reported by Gurjar and Cherian [130,138,139]. This synthetic route appeared to meet well with the need for suitably protected C12′, C13′, and C15′ hydroxyl groups possessing the correct configuration. Thus, the synthetic route was adopted by Song for the production of *tris*-TBS pentaenoate **49** (Figure 7) where the side chain alcohols were protected as *tert*-butyldimethylsilyl (TBS) ethers [104].

In the synthesis of the requisite fatty acid **49**, the aim was to synthesize first the C9′–C16′ *tris*-TBS aldehyde **47**. A more efficient synthetic route employed is outlined in Figure 8 where a known aldehyde, (*R*)-3-((tert-butyldimethylsilyl)oxy)butanal **50** available from ethyl (*S*)-3-hydroxy-*n*-butyrate in two steps [146], was converted to **51** by subjecting it to a Horner–Wadsworth–Emmons olefination. Reaction of **51** with AD-mix-α under catalytic asymmetric dihydroxylation [147,148], resulted in the formation of 3.8:1 mixture of diastereomers [149], with the expected and desired major product, a diol **52**. Boc protection of the diol by *bis*-TBS ether, and the subsequent sequence of reduction, oxidation, and Wittig olefination, produced the α, β-unsaturated ester **53**. Reduction of **53** yielded **54**, followed by chromatographic separation of the diastereomers and then oxidation, to form the requisite aldehyde **47**.

Employing modifications reported by Gurjar and Cherian afforded the C1′–C8′ phosphonate **46** [130]. Allylic alcohol **56** was synthesized in 4 steps from allyl alcohol **55** producing 25% overall yield (Figure 9). The diene ester **57** was formed from oxidation, followed by Wittig olefination of **56**, which was then a series of reactions involving reduction, oxidation, and Wittig olefination to produce triene ester **58**. Finally, a three-step sequence of deprotection, bromination, and phosphonate formation of triene ester **58** furnished the required phosphonate (2′*E*,4′*E*,6′*E*)-**46**.

#### 3.5.7. Completion of the First-Generation Synthesis of Mycolactone A/B

The total synthesis was completed by coupling the first-generation mycolactone core alcohol **26** (Figure 4) with the unsaturated fatty acids **49** (Figure 7) under Yamaguchi esterification conditions to form the protected mycolactone **59** in high yield (Figure 10). Subsequently, a global deprotection of the three silyl ethers and the cyclopentylidene acetal of **59** with HF•pyridine in acetonitrile led to mycolactone A/B in only 5–10% yield. A two-step sequence was then adopted using tetrabutylammonium fluoride (TBAF) followed by two successive treatments with an aqueous acetic acid solution. The three TBS groups on the highly unsaturated side chain of **59** were removed under standard conditions (TBAF/THF/rt), to yield the triol as an approximately 3:2 mixture of 4′*Z* and 4′*E* isomers in 81% yield. The cyclopentylidene ketal of **59** was then hydrolyzed by treatment with aqueous acetic acid [AcOH/H_2_O/THF (2:1:2)] at room temperature giving rise to a fully characterized synthetic mycolactone A/B approximately 3:2 mixture of 4′*Z* and 4′*E* isomers in 67% yield.

#### 3.5.8. Modular Total Syntheses of Mycolactone A/B

Saint-Auret et al. hypothesized that the C13–C14 σ-bond of the macrolactonic fragment **11** could be elaborated through a palladium-catalyzed Suzuki cross-coupling and that the tri-substituted C8–C9 double bond could arise from either a ring-closing metathesis or a palladium-catalyzed bromo-allylation reaction of an alkyne. On the other hand, the synthesis of the southern fragment **12** of mycolactone A/B was proposed to rely on two key steps, a palladium (0)-catalyzed and copper(I)-mediated Stille cross-coupling for the formation of the C7′-C8′ σ-bond as well as osmium-catalyzed asymmetric dihydroxylation reactions for the control of the C12′-, C13′-, and C15′-stereogenic centers (Figure 8). Following the hypothetical routes, they were able to synthesize the toxin albeit a mixture. However, this was inconsequential from the biological standpoint because the synthetic blueprint still allowed for late modification of the toxin [99,134,135].

## 4. Exploring Mycolactone as a Diagnostic Tool for Buruli Ulcer

*M. ulcerans* is unique among human pathogens in producing mycolactone, an attractive target for diagnosis and disease monitoring of BU disease. Histopathological studies have shown that in tissues, mycolactone is widely distributed compared to the causative organism [79]. Hong et al. have also established that mycolactone A/B appears to be biosynthetically restricted to *M. ulcerans*, homogeneously distributed within the infected tissue and shown to diffuse beyond the foci of primary infection hence an interesting target to be detected in circulating blood cells [73]. This understanding has resulted in several research groups exploring mycolactone as a biomarker in the diagnosis of BU. Several methods have been described for the detection of mycolactone in tissue samples, however, they are yet to be clinically deployed as diagnostic tools. In recent years, there has been increasing interest in exploring mycolactone as a unique biomarker for the diagnosis of BU. Some of the methods are described below.

### 4.1. Detection of Mycolactone Using Mass Spectrometry

Mass spectrometry analysis of acetone soluble lipid extracts from *M. ulcerans* sterile filtrates showed that mycolactone has been successfully detected with [M + Na^+^] at *m*/*z* 765.6 [79,150]. This was also confirmed using infected human tissue. Even though mass spectrometry data are often sensitive and specific, the technique is very expensive and sophisticated to be developed into point-of-care diagnostic tools for routine analysis, especially considering that disease is mostly endemic in low-resource areas. As a result, the use of spectrometry method has been limited to research purposes.

### 4.2. Immunological Assays

Immunological assays have great point-of-care potential because of their ability to be easily developed into POC devices; however, the development of an immunological mycolactone-specific test can be hampered by the lack of antibodies specific for the 743 Da polyketide. Despite this challenge, Warryn and colleagues recently developed mycolactone-specific monoclonal antibodies (mAbs) using a truncated synthetic mycolactone in which the lower side chain was replaced by a linker molecule [151] (Figure 9).

Subsequently, they developed competitive antigen capture enzyme-linked immunosorbent assays (ELISAs) using selected mAbs. The generated assay displayed high sensitivity to mycolactone in the low nanogram concentration range [152,153]. This achievement notwithstanding, the researchers are yet to translate the mycolactone-targeting ELISAs into POC lateral flow immunoassays (LFIAs) for deployment in resource-poor settings. This is because mycolactone is light-sensitive and amphiphilic and tends to either self-aggregate or form complexes with serum proteins or other substrates, thereby limiting the amount of mycolactone available in clinical samples [154]. Nonetheless, by relying on the first mycolactone-specific monoclonal antibodies and biotinylated mycolactone ELISA as reported by Warryn et al. [152], efforts were made to convert it into a lateral flow immunoassay (LFIA). The prototypical LFIA was employed by Sakakibara et al. in a case report of a Japanese patient for diagnosis [155]. However, the evaluation of the antigen-specific method was limited to a single case report, therefore its diagnostic performance could not be established.

### 4.3. Diagnosis of Buruli Ulcer with RNA Aptamers

As discussed above, there is considerable difficulty in the body’s ability to produce antibodies against mycolactone, partly due to the lipid nature of the molecule. To address this, Sakyi et al. sought to provide a potential alternative method to the mAbs by exploiting nucleic acid molecules with high affinity to its target called aptamers [156]. Aptamers have been selected that possess high affinity and specificity for mycolactone and hence have the potential of being developed for use as an *M*. *ulcerans* detection assay [157]. RNA aptamers that bind to mycolactone were isolated and evaluated in a pilot study. The aptamer-based assay was used in a case–control study using swab samples from 41 suspected BU patients and had a sensitivity of 50%, which is comparable to that of microscopy and culture [158,159], and a specificity of 100% [156] comparable to that of the *IS2404* PCR [160,161]. Compared to routine diagnostic tests, aptamer-based detection assay has better stability under varied circumstances and can be used repetitively. This is promising for the development of the aptamers as recognition molecules for diagnosing BU hence, they serve as detection molecules for the development of better diagnostic assays [162]. Even though the aptamer assay holds promise for BU diagnosis, it was low on sensitivity (50%) despite being highly specific (100%). Additionally, the method was limited by the absence of a speedy readout, and was time-consuming thus, requiring many hours to finish an ELONA, and there was also the major issue of antigenic cross-reactivity. These drawbacks can influence the ultimate performance of the aptamer assaying method.

### 4.4. Fluorescent-Thin Layer Chromatography (f-TLC)

Mycolactone can be detected using classical thin layer chromatography (TLC) to be a distinct light yellow UV-active lipid with a retention factor (Rf) of 0.23 using a chloroform–methanol–water (90:10:1, vol/vol) elution mixture [57,60,163]. Later, Sarfo et al. demonstrated that mycolactone could be detected in infected human tissue on TLC with a similar Rf value. This was consistent with previous literature data from TLC analysis of acetone soluble lipid extracts from *M. ulcerans* sterile filtrates [57,164]. Subsequently, the fluoresecent-thin layer chromatography (f-TLC) method was developed by Kishi and colleagues [165]. The method relies on the fluorescence detection of mycolactone by taking advantage of the intrinsic affinity of boronic acids for 1,2-cis and/or 1,3-cis diol moieties [166]. The 1,3-diol units of mycolactone A/B were chemically derivatized using 2-naphthylboronic acid (BA) as a fluorogenic chemosensor to form two cyclic boronates [165]. Irradiation at 365 nm UV light led to an enhanced fluorescence emission intensity from the pentaenoate, enhanced by the C13, C15-cyclic boronate (Figure 10). This method allows the detection of as low as 2 ng of mycolactone within a considerably reduced background and it is specific for mycolactones A/B, C, and D. Mycolactones E and F do not yield fluorescent spots or bands. Furthermore, the method has been validated in a mouse footpad model of *M. ulcerans* infection [167]. The method has subsequently been confirmed in infected clinical human skin samples; first in a small sample size [168]. With a detection rate of 73%, TLC is superior to microscopy (30–60%) or culture (35–60%) and comparable to histology (82%), but inferior to PCR (92–98%) [169,170].

Later, the f-TLC was evaluated at the district hospital level using a larger sample size of 449 suspected cases. The method returned a sensitivity of 66.3%, a specificity of 88.5%, and was 82.2% accurate when compared with PCR [171]. Recently, an alternative boronic acid, (9,9-Diphenyl-9H-fluoren-4-yl)boronic acid (BA18), has been identified as a potential replacement of the original boronic acid [172]. The f-TLC method has some important advantages over existing methods including being robust, low-cost, user-friendly, and requires simple instrumentation while still giving results in a short time. Despite the advantages, it has drawbacks including background interferences from co-extracted lipids that co-elute with the target analyte thus making the method susceptible to user interpretation. In addition, differences in test conditions can result in different readings and interpretations of the same test [165,168,171]. Consequently, these limitations have hindered the point-of-care applications and clinical utility of the method particularly in resource-limited endemic settings.

## 5. Mycolactone: The Immunosuppressor, the Analgesic, and the Cytotoxic Toxin

Extensive ulcerative lesions observed in patients suffering from BU can be attributed to the secretion of mycolactone, the toxin produced by *M. ulcerans* [57]. Additionally, the toxin is widely accepted to be responsible for the painlessness that is characteristic of lesions caused by *M. ulcerans*, especially in early stages of the disease where there are no concerns of co-infection [8,173,174]. Painlessness has been demonstrated in mice models where mycolactone was observed to induce long-lasting hypoesthesia [62,66]. In recent years, the painless cytotoxin has also been demonstrated to be a natural immunosuppressor in cellular models [80,82,175,176,177]. Considering the role of mycolactone, various researchers have made significant contributions aimed at deciphering the molecular and pathogenic mechanisms underlying the action of mycolactone-mediated ulcerative, immunosuppressive, and analgesic properties [58,176,178,179,180,181].

### 5.1. Cellular Targets Responsible for the Mechanism of Action of Mycolactone

Several cellular targets have been proposed by various studies to account for the numerous biological effects of mycolactone such as skin ulceration, host immunomodulation, and analgesia. A number of internationally acclaimed scientists have studied mycolactone and proposed that its inhibitory mechanisms are accounted for by (i) inhibition of the WASP and related neural N-WASP [178,182], (ii) inhibition of Sec61-dependent translocation of proteins into the endoplasmic reticulum [176,181], and (iii) inhibition of angiotensin II type 2 receptor (AT2R) as illustrated in Figure 11 [58,62,183]. There is also the proposition that mycolactone acts as an inhibitor of the mechanistic Target of Rapamycin (mTOR) which is attributed to the induction of apoptosis in mammalian cells [85].

#### 5.1.1. Wiskott–Aldrich Syndrome Protein (WASP) Inhibition

The WASP and N-WASP have been identified as molecular targets of mycolactone. The WASP as mycolactone target was first reported by the Demangel group [178,182]. WASP and N-WASP belong to a family of scaffold proteins that mediate the dynamic remodeling of the actin cytoskeleton [182]. In cell-free assays of actin polymerization, the mechanism involves mycolactone being able to mimic the endogenous GTPase CDC42, and thus able to hijack the WASP and N-WASP. It does so by binding into WASP and N-WASP and activating it with a 100-fold greater affinity than it binds to the natural activator CDC42. This results in alterations in actin dynamics, leading to defective cell adhesion, uncontrolled directional migration, and eventually leading to apoptosis. Thus, the mycolactone-driven destruction of cutaneous tissues and analgesia was attributed to the hyperactivation of neural N-WASP (Figure 11). The neural form (N-WASPs) particularly plays a key role in the ulcerative effects of mycolactone and as a result, it is more expressed than its hematopoietic homolog, the WASP [62,182].

#### 5.1.2. Inhibition of Sec61

Proteins for the endocytic system originate by translation in the cytoplasm, and typically enter the endoplasmic reticulum (ER) cotranslationally. These nascent polypeptides usually enter the ER via the Sec61 translocon and engage the glycosylation process [184]. Sec61 translocation is responsible for the transport of about 30 to 50% of all mammalian proteins, including but not limited to circulating inflammatory mediators, immune mediators, and proteins involved in lipid metabolism, coagulation, and tissue remodeling [183,185,186]. Just as with the N-WASP, the Sec61 translocon was also identified by Simmonds et al. as the second major primary cellular target of mycolactone (Figure 11). The Sec61 translocon is responsible for the cotranslational translocation of newly synthesized proteins through the endoplasmic reticulum (ER) membrane. Therefore, its inhibition is thought to play a critical role in the expression of immune system mediator proteins. Cotransin, decatransin, apratoxin, ipomoeassin, cyclotriazadisulfonamide, and eeyarestatin are a few structurally different natural and synthesized small compounds that bind to Sec61 in a substrate-specific, selective way and prevent protein translocation [187]. Considering that the Sec61 translocon is a primary target of mycolactone, it was hypothesized that cellular effects including cytotoxicity of mycolactone could be attributed to its inhibitory effect on the translocon in the cytosol. Following this, Hall et al. demonstrated the ability of mycolactone to specifically block the Sec61 translocon and hinder protein translocation into the ER, thus causing the ubiquitin–proteasome system to degrade the proteins in the cytosol. This potentially contributes to tissue damage, immunosuppression, and ultimately, cell death [176,188]. This is corroborated by the findings of McKenna and colleagues who demonstrated using cell-free experimental assays that mycolactone selectively influences the step of cotranslational translocation of secreted transmembrane proteins (TMPs) into the ER [181]. In another study, Kawashima et al. screened host factors and investigated the mycolactone-induced cell death pathway in human premonocytic THP-1 cells using a genome-scale lenti-CRISPR mutagenesis test. This study was noteworthy since it revealed that SEC61A1, the α-subunit of the Sec61 translocon complex, is a crucial mediator of mycolactone-induced cell death [189]. Despite these findings, the mechanism behind the linkage between mycolactone translocation blockage and apoptosis-induced cell death remained unclear. Gérard et al. used electron cryo-microscopy to analyze translocon and identify the structure of a mycolactone-inhibited mammalian Sec61 in an attempt to better understand the molecular mechanism of action [190]. The first thing they demonstrated was that mycolactone wedges open the cystolic side of the Sec61α lateral gate. This stabilizes a poised and permissive conformation for post-translational translocation, which is remarkably similar to the conformation stabilized by Sec62/63. Furthermore, they demonstrated that mycolactone concurrently inhibits the path of the signal peptide while holding the translocon in a conformation that is optimal for substrate engagement signals [190]. Given that mycolactone-inhibited Sec61 is unable to transport substrate proteins effectively, it is possible for transmembrane domains and signal peptides to pass through the mycolactone-occupied regions. This reveals novel facets of translocon dynamics and activity and establishes the foundation for understanding the molecular mechanism of Sec61 inhibitors.

Additionally, considering that mycolactone is amphiphilic in nature, it allows for significant interactions with lipophilic environments. However, it is unclear how specific these connections are and how membranes contribute to the pathogenicity of the toxin. Consequently, da Hora and colleagues used enhanced free-energy investigations and molecular dynamics simulations to characterize the association and penetration of mycolactone via the plasma membranes (PMs) and endoplasmic reticulum (ER) of mammals. They demonstrated that lipid content influences both membrane association and mycolactone penetration. The ER membrane keeps its bent c-shape conformation as the two polar tails of the toxin spread across the bilayer in the plasma membranes (PMs). In the PM, this is the most common penetration mechanism [190]. The cause is most likely the cholesterol in the PM, which has a propensity to induce stiffness and thicken the bilayer. This suggests that during penetration, it is nearly hard for the poison to alter the PM while preserving the polar contacts with the water molecules and lipid head groups. Likewise, the free-energy calculations demonstrate that mycolactone has a greater affinity for the ER membrane than PM due to the presence of saturated lipids and cholesterol in the latter [191]. On the other hand, investigations using the same molecular dynamics simulations indicate that mycolactone isomer B has a substantial affinity for the ER membrane, which raises the local concentration around Sec61 and may increase the effective affinity for the translocon [192].

#### 5.1.3. Inhibition of Angiotensin II Type 2 Receptor (AT2R)

In addition to the inhibition of Sec61 by mycolactone, Marsollier et al. identified angiotensin II type 2 receptor (AT2R) as the main receptor involved in mycolactone-mediated analgesia [62]. They demonstrated the ability of mycolactone to exert an analgesic effect through the inhibition of angiotensin II type 2 receptor (AT2R). Potassium (K+) channels form a large family of hyperpolarizing channels, as such, they are expressed in every cell of the organisms. They are involved in cellular mechanisms including apoptosis, vasodilation, anesthesia, pain, and neuroprotection and they achieve this by producing background currents that oppose membrane depolarization and cell excitability [193]. Mycolactone therefore is able to bind to angiotensin II type 2 receptors (AT2Rs) on neurons, leading to potassium-dependent hyperpolarization. Marion et al. were able to show that mycolactone activates AT2R in neurons, with the release of phospholipase A2-mediated arachidonic acid (PLA_2_), the generation of prostaglandin E2 from AA by cyclooxygenase-1 (PGE_2_) and the subsequent release of potassium through TRAAK channels (Figure 11) [62]. Notably, drugs known to activate potassium channels display antinociceptive properties as they induce hyperpolarization, thus decreasing neuron sensitization [194]. The sustained hyperpolarization induced in sensory neurons was proposed to mediate the analgesic properties of mycolactone. This has been validated in mice that were inoculated with mycolactone being less sensitive to a pain stimulus using the Hargreaves plantar pain test and this effect was dependent on the type-2 angiotensin II receptors (AT2R) [195,196]. In a separate study, Anand and colleagues sought to decipher the mechanisms responsible for mycolactone-induced analgesic effects in rat and human primary dorsal root ganglion (DRG) sensory neurons [179]. They observed significant neurite degeneration in rat and DRG sensory neurons after 24 h of exposure to a 100 nM concentration of mycolactone. These findings are consistent with previous in vivo studies by En et al. using mice models where mycolactone was thought to induce hypoesthesia with an extremely long-lasting effect leading to nerve destruction at late stages of the disease [65,66,116]. Additionally, the functional effects of known ligands of AT2R, including agonists or antagonists such as angiotensin II, C21, and EMA 401 in cultured human and rat DRG neurons were investigated by Anand et al. It was demonstrated that only mycolactone was able to trigger hyperpolarization of DRG neurons [197,198]. More interestingly, the AT2R antagonists, EMA 401 or C21, were unable to inhibit hyperpolarization that was triggered by the toxin. This observation confirms the specificity of mycolactone/AT2R interactions and is an indication of the lack of involvement of the AT2R in ML-mediated neurotoxicity [179]. Consequently, it has been proposed that the biological mechanisms of mycolactone-induced cell death entail the hijacking of various signaling pathways. For example, the mTOR pathway and the system requiring Sec61-dependent protein translocation into the ER in which the mycolactone - disrupts the mTOR and Sec61 pathways in DRG neurons [85,185].

#### 5.1.4. Inhibition of mTOR

Mycolactone has been shown by Bieri et al. to bind to and inhibit the mTOR signaling pathway in the body by using L929 fibroblasts as a model [85]. The mTOR forms two structurally and functionally distinct complexes called the mammalian target of rapamycin complex 1 (mTORC1) and mammalian target of rapamycin complex 2 (mTORC2), which together are responsible for the regulation and control of cell growth, metabolism, cell proliferation, and survival of several cell processes including apoptosis by participating in multiple signaling pathways in the body [199]. When mTOR (mTORC1 and mTORC2) complexes are exposed to mycolactone, the mTORC2 complex is particularly inhibited by the toxin. This prevents phosphorylation of the serine/threonine protein which ultimately inhibits kinase Akt activity. Once Akt is inactivated, it leads to the dephosphorylation and activation of the Akt-targeted transcription factor FoxO_3_. Subsequent upregulation of the FoxO_3_ target gene BCL2L11 (Bim) increases expression of the pro-apoptotic regulator Bim, driving mycolactone-treated mammalian cells into apoptosis through the mTORC2-Akt-FoxO_3_ axis [85]. Signal transduction requires the pro-optotic protein Bim, which is encoded by the BCL2L11 gene. It has been demonstrated that experimentally infected Bim mutant animals can inhibit M. ulcerans growth and do not form necrotic BU lesions [85]. This conclusion is supported by the observation that human ulcerative forms of Buruli ulcer are linked to genetic variations in BCL2L11, indicating that apoptosis regulation is a critical factor in the disease pathogenicity [200].

### 5.2. Structure–Activity Relationship (SAR) Studies of Mycolactone

*M. ulcerans* is an extremely slow-growing cytotoxic environmental pathogen that must be cultured in higher safety laboratories. It is therefore difficult to obtain it by cultivation of *M. ulcerans*; meanwhile, mycolactone is required in sufficient quantities for a variety of studies including the structure–activity relationship (SAR). Fortunately, since 2002, various research groups pioneered by the Kishi group together with Song et al. [104], Yin et al. [122], Feyen et al. [201], and Chany et al. [202] have made tremendous strides in providing synthetic routes (involving not less than 30 elaborate steps) for the total synthesis of mycolactone and its variants. Details of the contributions were discussed earlier. However, the complexity of chemical schemes and the huge time and resources required for the total synthesis of the complex macrolide coupled with very low yields made large-scale production for clinical applications challenging [38,99,103]. Nonetheless, enough have been provided for the investigation of structure–activity relationship (SAR) studies [38] and cellular assays [97]. For instance, the phenomenal work of the Kishi group in the total synthesis of mycolactone A/B and the other naturally occurring variants fundamentally set the stage for structure–activity relationship (SAR) studies. It was revealed that the lactone core was essentially conserved in all the natural variants with variations only in the nature of the southern chain (Figure 12). This was the first indication that the conserved lactone core could be essential for biological activity of the different natural analogues of mycolactone. This finding was fundamental in the design and synthesis of novel analogues with varying southern chains for further structure–activity relationship studies. For example, Kishi’s group synthesized mycolactone analogues **56a** and **56b** by employing chain extension approaches of the polyketide southern side chain (Figure 12). They employed **56a** as a precursor to synthesize other mycolactone conjugates including the amide **56b**, which was shown to be cytotoxic (30 nM) against L929 fibroblasts compared to mycolactone A/B which gave 10 nM (Figure 12). This was a clear indication that elongating the southern chain was well tolerated considering the comparable activities of mycolactone A/B and the **56b** [63].

Thereafter, other research groups employed other simplification strategies to generate a library of mycolactone analogues in order to investigate further the minimal structural determinants of biological activity [84,183,203]. For instance, Altmann and Pluschke investigated different modifications of the C-linked upper side (comprising C12–C20) and the lower C5-O-linked polyunsaturated acyl side chain. The biological activity of the various analogues was tested against murine L929 fibroblast cell lines measured by flow cytometry and the LC_50_ (the concentration of analogue for which half of the cells were killed) were recorded [84,128,151].

The first category of analogues generated by Altmann and Pluschke involved esterification of the conserved lactone core together with the northern side chain with various esters via the C5-O-linkage [84,128]. For analogue **57a**, where the southern chain was absent but with a free hydroxyl group at position C5, LC_50_ > 5000 nM was obtained when it was tested against murine L929 fibroblast cell lines thus indicating inactivity. Likewise, the analogue **57b** which was esterified with an acetate residue at C5-O did not also show any activity at working concentration (LC_50_ >> 5000 nM). However, when a sorbate ester was employed in the esterification to obtain **57c**, residual cytotoxicity was measured against the L929 fibroblast cell lines (LC_50_ = 3426 nM). Furthermore, analogue **57d**, devoid of all hydroxyl groups at positions C12′, C13′, and C15′ on the southern side chain as compared to the mycolactone A/B and it only showed minor activity (LC_50_ = 4550 nM) (Figure 13).

The second category involves an incomplete mycolactone A/B with only the lactone ring being conserved whereas both the northern and southern chains were varied. The northern chain was truncated at C13 (thus, with an isopropyl substituent at C11) while the southern chain was either completely removed and replaced with a hydroxyl group at C5 (**58a**) or esterified with sorbate ester to obtain **58b**. In both analogues, there was either no or very reduced cytopathic effect.

The third category was designed by exploring polar groups as substituents at C20, starting with a hydroxyl group for the analogue **59a**. The introduction of the hydroxyl group had a significant effect on cytotoxicity giving an LC_50_ of 15 nM for **59a**, similar to that of natural mycolactone A/B (12 nM), when they were both tested against L929 fibroblast cell lines [84]. Upon derivatization of the hydroxyl analogue via extension modules to the acetate analogue **59b**, and a bulkier carbamate analogue **59c**, LC_50_ of 45 nM and 50 nM, respectively, were obtained.

Thus, the flow cytometry data of **59a**, **59b**, and **59c** (with LC_50_ ranging from 15 nM to 50 nM) indicated that functionalizing of the upper side chain was relatively tolerant. Again, from the cytopathic effect data of **58a** and **58b** together with that **59a**, **59b**, and **59c**, it could also be concluded that the presence of the southern chain was important for cytotoxicity of mycolactone analogues [84].

Following these successes in the Altmann and Pluschke modifications, Blanchard et al. proceeded to investigate further the influence of the southern chain by generating a diverse library of C8-desmethylmycolactone analogues (i.e., mycolactone analogues lacking a methyl substituent at C8 of the lactone core) (Figure 14) [183,202]. They prepared nine novel C8-desmethylmycolactone analogues with varying northern and southern chains and investigated their cytopathicity by measuring the cytopathic effect (CPE). The CPE is the concentration of mycolactone analogue for which 90% of the cells round up. The CPE was determined based on cytopathic assays on L929 fibroblasts in comparison with natural mycolactone leading to the establishment of further structure–activity relationships [188].

In the first series of five C8-desmethylmycolactone analogues, the absolute configuration at C12′, C13′, and C15′ stereocenters on the southern side chain were investigated. The most potent analogues were **60a** (possessing the configuration of the natural mycolactone A/B around C12′, C13′, and C15′) and **60b** (where C15′ configuration is opposite that of the natural analogue). They induced 100% cell rounding within 48 h at 10 μM concentration. The natural mycolactone A/B, on the other hand, induced 90% cell rounding at a minimum concentration of 40 nM within 24 h [90]. The results obtained with **60b**, where 100% cell rounding was achieved suggesting that the configuration at C15′ was less crucial. In **60c**, where the three stereocenters C12′, C13′, and C15′ were inverted, the cytopathic effect for the *epi*-stereotriad analogue (**60c**) dropped considerably to 10% at 10 μM concentration. These findings confirm the essentiality of the stereochemistry of the C12′ and C15′-hydroxyl groups because the deoxy-C12′ and deoxy-C15′variants **60d** and **60e** resulted in a drastic reduction in activity. For instance, only 49% and 40% cell rounding were achieved at a concentration of 10 μM for **60d** and **60e** within 48 h, respectively. On the other hand, 100% cell rounding was achieved within 48 h under the same concentration for compounds **60a** and **60b** in which the hydroxyl groups were present at C12′ and C15′, thus reinforcing the importance of the C12′ and C15′ hydroxyl groups (Figure 14) [202].

Secondly, the next four other analogues of the C8-desmethyl analogues were **61**, **62**, **63a**, and **63b; 61** comprised the C8-desmethyl lactone core with a northern side chain but without a polyunsaturated southern chain while **62** comprised the polyunsaturated southern side chain alone. Furthermore, **63a** is comprised of the C8-desmethyl lactone core with neither the northern side chain nor the southern polyunsaturated side chain, and **63b** contains the southern polyunsaturated side chain without the northern side chain. When the four were tested for cytopathicity, the results showed that neither the polyunsaturated southern side chain **62** alone (CPE = 5%) nor the C8-desmethyl lactone **63a** alone (CPE = 10%) were active (Figure 14). On the other hand, the analogues **61** and **63b**, which contained the C8-desmethyl lactone core (C1–C11) with either the northern side chain (C13–C20) or the southern polyunsaturated side (C1′–C16′), returned some cytopathic potential at 10 μM, albeit, low at 27% and 53% for **61** and **63b**, respectively. These results suggest that the combination of the northern side chain, central C8-desmethyl lactone core, and southern polyunsaturated side chain are essential for the potency mycolactone and its analogues.

### 5.3. Therapeutic Potential of Mycolactone Analogues

Following the synthesis of various analogues of mycolactone using elaborate synthetic approaches [97,128,202], biological effects were further elucidated in several structure–activity studies [84]. The key finding from the structure–activity relationship (SAR) studies was the revelation that pruning the side chains of the structure of mycolactone in any way was unfavorable for biological activity. It was particularly observed that an intact southern polyketide side chain was essential for cytopathic activity whereas a mycolactone A/B core structure devoid of both the northern side chain and the southern polyketide chains yielded a biologically inert analogue [90,202]. For instance, Chany et al. demonstrated that a group of simplified analogues of mycolactone without the northern polyketide chain bind to WASP/N-WASP with a binding potency equivalent to that of the natural toxin [203]. In the mechanistic process, various cellular targets such as WASP and N-WASP, the Sec61 translocon, angiotensin II type 2 receptor (AT2R), and mTOR have been identified as molecular targets of mycolactone A/B and attributed to its immunosuppressive, analgesic, and cytotoxic properties.

#### 5.3.1. Analgesic Effects of Mycolactone Analogues as Potential Pain Killers

According to the International Association for the Study of Pain (IASP), pain is defined as “an unpleasant sensory and emotional experience associated with, or resembling that associated with, actual or potential tissue damage” [204]. Albert Schweitzer (1931) succinctly stated that “pain is a more terrible lord of mankind than even death itself” [205]. Pain is a symptom of many medical conditions and is currently reported as the most common medical problem for which many patients tend to seek medical attention [206]. The Institutes of Medicine and the American Pain Society estimate that pain affects more than 100 million adults in the United States and costs about USD 635 billion each year in medical treatment and lost productivity [206,207]. Therefore, the relief of pain remains an important therapeutic goal of many researchers by relying heavily on therapeutic agents with analgesic properties. The commonly available drugs for the management of pain include non-narcotic analgesics (acetaminophen and aspirin-NSAID), narcotic analgesics (opioids), non-steroidal anti-inflammatory drugs (NSAIDs), and thermal agents. In recent years, other medicines such as anticonvulsants, antidepressants, and selective cyclooxygenase 2 (COX2) inhibitors have been added [208]. Even though these Food and Drug Administration (FDA)-approved drugs offer relief to pain sufferers, they are often associated with poor tolerability, limited effectiveness, significant and unfavorable side effects, concerns over long-term safety and abuse potential, as well as inconvenience of use [208,209,210]. Owing to these challenges, efforts are constantly channeled at finding novel analgesic agents with safer profiles and efficacy. Scientists have explored the characteristic painlessness associated with Buruli ulcer lesions, which is widely attributed to the secretion of mycolactone to generate drug candidates with the therapeutic potential in treating pain [183,197,211].

Following the in vitro and in vivo studies of the mycolactone-like analogues, the variant of mycolactone A/B devoid of north chain and core C8-methyl (**63b**) induced 53% of cell rounding at 10 μM compared to 90% cell rounding as observed in the natural mycolactone A/B [90,202]. Even though the C8-desmethyl analogue **63b** was less potent than natural mycolactone, it displayed enhanced binding to AT2R and inhibited inflammatory cytokines production [183]. Moreover, the truncated version retained the capacity to block cytokine production by neutrophils, macrophages, and lymphocytes, although with less potency. From a clinical perspective, **63b** offers several advantages over the natural molecule. In addition to being easier to chemically synthesize, its smaller size (572 versus 743 daltons) would allow a better distribution throughout the fluids and tissues of the body. Second, cellular assays showed that **63b** retains the desirable (immunomodulatory and AT2R binding) properties of mycolactone relatively better than its undesirable (cytotoxic) ones [183]. Therefore, **63b** could be engineered as a potential painkiller as an alternative to full mycolactone. From the perspective of using mycolactone (or derivatives) as an analgesic, it is important to emphasize that induced toxicity was observed only for high concentrations (>5 µg) or high incubation times (>24 h) [197].

#### 5.3.2. Mycolactone Analogues as Potent Immunosuppressive Agent

Mycolactone-like molecules were investigated as drug candidates against chronic skin inflammation [183]. Comparatively, immunomodulatory activity was retained in **63b** when compared to natural mycolactone A/B (**1**, IC_50_ of 1.5 mM) [183]. Furthermore, Guénin-Macé and co-workers attempted to delineate the immunosuppression and analgesia from cytotoxicity [183]. They were inspired because similar strategies have previously been successfully developed and implemented for other toxins such as conotoxins, where slight modifications of their structures made them devoid of adverse effects [212,213] and they found that mycolactone variant **63b** displayed optimal immune suppression, without cytotoxicity. In fact, they demonstrated that non-cytotoxic dose levels of **63b** could bind to angiotensin II type 2 receptors as well as mycolactone does and therefore effectively inhibited inflammatory cytokine responses of human primary cells in vitro. The added advantage was that **63b** was considerably less toxic than mycolactone in human primary dermal fibroblasts modeling ulcerative activity. This means they have great prospects as immunosuppressants.

#### 5.3.3. Other Therapeutic Applications of Mycolactone Derivatives

Mycolactone A/B analogues such as **63b** with the southern polyketide side chain have shown the capacity to activate WASP/N-WASP in vitro and induced cytopathic effects. Biosynthetic studies have demonstrated that mycolactone-like compounds with a truncated/modified structure lost cytotoxicity compared to whole mycolactone [84,97,202]. Generally, various structural analogues of mycolactone have been shown to significantly exhibit different immunosuppressive activities [97], with the loss of hydroxyl groups reducing the cytotoxicity of mycolactone [84]. This is evidenced and consistent with displayed reduced toxicity along with a reduced immunosuppressive effect observed in mycolactone C, a natural derivative of mycolactone A/B lacking a hydroxyl group, as well as the synthetic structural variant devoid of all the hydroxyl groups in the lower side chain. Since mycolactone is the sole pathologic factor of *M. ulcerans*, these findings on the cytotoxic and immunosuppressive effects imply that the inhibition of mycolactone activity or its SEC61A1 could be a novel therapeutic modality to eliminate the harmful effects of mycolactone. In recent studies, researchers have sought to exploit this phenomenon of the toxin and its analogues for their anti-viral properties [214]. This is because recent screen-based studies revealed that Sec61 is required for viral replications in dengue, HIV, and Zika viruses [214,215]. For instance, the translocon inhibition of Zika virus by mycolactone prevented viral replication, the formation of the characteristic ER-derived vacuoles, and the ensuing cell death [214]. It was found that the treatment of cells with mycolactone at 2 h resulted in complete inhibition and vacuolization. Mycolactone exhibits this mode of action by directly targeting the subunit of the Sec61 translocon, therefore, cells expressing a mutant Sec61a resistant to the effects of mycolactone (Sec61 mut) were utilized [185]. The virus was not inhibited by mycolactone in cells expressing mutant Sec61, which is an indication that a Sec61 blockade protects against ZIKV infection. For the treatment of influenza virus, mycolactone targeted viral envelope proteins just as well as endogenous targets, with specificity for type I/II but not type III TMPs for its cytopathic effects [214]. Also Sec61 blockers have been exploited for their therapeutic potential for the treatment of plasma cell malignancy multiple myeloma (MM) which still remains an incurable oncology disease [188]. Mycolactone has been demonstrated to effectively decrease the synthesis of various type I/II TMP receptors, including CD138, and immunoglobulins in MM cell lines. A characteristic of MM, CD138 promotes growth factor signaling, which enables malignant cells to survive in the bone marrow [216]. Additionally, mycolactone therapy reduced the expression of CD40, which increases the production of IL-6, and the pro-survival IL-6 receptor in MM cells. In MM cell lines and tumors derived from MM patients, mycolactone later caused a pro-apoptotic ER stress response. This demonstrated the potential of Sec61 inhibitors as an anti-cancer treatment for MM and served as proof of concept [216].

## 6. Conclusions and Future Perspectives

The macrolide toxin, mycolactone A/B, is an essential virulence factor of *Mycobacterium ulcerans* and plays a central role in the pathogenesis of the disease hence classified as the causative toxin of Buruli ulcer. Mycolactone has been demonstrated to induce necrosis and ulceration by its cytotoxic and immunosuppressive properties. Mycolactone A/B in particular displays potent inhibition of interleukin production. Such effects of mycolactone are associated with a lack of wound healing and the eventual formation of large symptomatic ulcerations. In view of these roles, a better understanding of the chemistry and biosynthesis of the toxins is necessary in order to appreciate the delicate interplay between biological effects and chemical structures. To explore the biological potencies of these toxins and help develop an understanding of their mode of action, increasingly larger quantities of mycolactone A/B are required. However, this has been seriously hindered by the slow growth of the host mycobacterium and the small quantities of the toxin that are available from laboratory-scale culturing efforts, respectively. For these reasons, mycolactone has attracted considerable attention from the synthetic community with the aim of providing efficient and scalable synthetic routes to afford pure synthetic mycolactone for purposes of research. This review highlights the mycolactone A/B chemistry, biosynthetic, and synthetic pathways. Furthermore, the interaction of mycolactone and its derivatives with the cellular targets including WASP/N-WASP, the Sec61 translocon, angiotensin II type 2 receptor (AT2R), and inhibition of mTOR accounted for their immunosuppressive, analgesic and cytotoxic properties as well as the potential of being used for the treatment of inflammatory pain and inflammatory diseases. There is also the great therapeutic potential of Sec61 blockers because they showed broad spectrum anti-viral activities by acting both on the formation of essential viral proteins and by inhibiting viral entry in the host through different mechanisms. We believe that Sec61 blockers have the potential to be novel and powerful anti-viral tools in the near future.

## Data Availability

No new data were created or analyzed in this study. Data sharing is not applicable to this article.

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
