# Peer review of "Exploring Mycolactone—The Unique Causative Toxin of Buruli Ulcer: Biosynthetic, Synthetic Pathways, Biomarker for Diagnosis, and Therapeutic Potential"

_toxins, 2024, doi:10.3390/toxins16120528_

Round 1
Reviewer 1 Report
Comments and Suggestions for Authors
The manuscript reviews the most relevant information about mechanism of action, different synthetic pathways and therapeutic potential of mycolactones A/B and their SAR-generated analogs. The manuscript also describe the etymology of Buruli ulcer and highlights the need of further studies to improve the challenging clinical practice associated with these infections.
Author Response
Reviewer 1
The manuscript reviews the most relevant information about mechanism of action, different synthetic pathways and therapeutic potential of mycolactones A/B and their SAR-generated analogs. The manuscript also describes the etymology of Buruli ulcer and highlights the need for further studies to improve the challenging clinical practice associated with these infections.
We are grateful for the comments by the reviewer.

Reviewer 2 Report
Comments and Suggestions for Authors
This paper is a review of the chemistry and biological effects of mycolactone, a bacterial toxin produced by the causative agent of Buruli ulcer, Mycobacterium ulcerans. The paper covers the background on the disease, the chemistry and synthesis of mycolactone and its analogues, biological mechanism of action, the use of mycolactone in diagnosis and its potential as an analgesic and anti-inflammatory agent. While some aspects are clear and well written, overall the review seems quite dated. The main focus is on the chemical synthesis of mycolactone and while this work was groundbreaking, the most recent paper cited is from 2017 and much of the ground has been covered in previous reviews (eg Gehringer and Altmann, Beilstein J Org Chem 2017, 13, 1596). In other areas, such as mechanism of action, diagnostics and therapeutics there are many omissions of relevant recent papers and there is a tendency to just summarise reports and ignore discrepancies between them when a critical analysis would be more valuable to the reader.
Minor concerns
1. Figure 1: Image described as Plague should read Plaque.
2. There is a lot of unnecessary repetition eg line 51 is almost identical to line 57. This is a long review and could be cut down considerably by removing such repeats.
3. Line 78 and 89: should be Speke not Specke.
4. Mycolactone is sometimes referred to in the singular and sometimes in the plural eg paragraph starting on line 185 uses both interchangeably. This makes sense where different natural or synthetic analogues are used but this is not the case for many of the reports cited.
5. Line 330-31: No citation for Gurjar or Dai.
6. Line 614: reference 182 is wrong.
7. Line 726. The author’s state that 100% rounding at a concentration of 10µM at 48hr is comparable to a 90% rounding at 40nM at 24hr. The values are not comparable and the two assays are totally different and should not be compared in this way.
8. Line 769: binding potency to what?
Major concerns
1. Diagnostics: the section on the use of mycolactone for diagnosis discusses TLC and aptamer methods but does not cover the recent advances in antibody dependent detection using either ELISA or lateral flow assays (Warryn et al, 2020 https://doi.org/10.1371/journal.pntd.0008357, Warryn et al, 2021, https://doi.org/10.4049/jimmunol.2001232, Sakakibara et al, 2024 https//doi: 10.1016/j.jctube.2024.100469). These should be fully discussed in the review.
2. Mechanism of action: The authors do not mention the work described in ref 84 proposes a fourth target of mycolactone, mTOR. This should also be discussed in this section.
3. Mechanism of action: The authors give the impression that Sec61 inhibition is solely responsible for the immunosuppressive and anti-inflammatory actions of mycolactone. This is misleading as many of the changes in cell morphology and adhesion properties can be also attributed to this mechanism. Furthermore, the identification of Sec61 as a target using either chemical (ref 180) or CRISPR (Kawashima et al 2022 https://doi.org/10.1371/journal.pntd.0010672) based screens shows that Sec61 inhibition is a major factor in mycolactone driven cytotoxicity. To date, Sec61 is also the only target for which an interaction with mycolactone has been backed up with structural data (Gerard et al, 2020 https://doi: 10.1016/j.molcel.2020.06.013, Itskanov et al, 2023 doi: 10.1038/s41589-023-01337-y). These are backed up by molecular dynamics studies from the Swanson lab which demonstrate a preference for ER membranes and suggest the differences between mycolactone A and B interaction with membranes and the translocon could explain their different potencies (Nguyen et al, 2023 https://doi: 10.3390/toxins15080486 and da Hora et al, 2022 htpps://doi: 10.1016/j.bpj.2022.10.019). These are highly relevant studies that should be included in the review.
4. Analgesic effects of mycolactone. The authors cite a paper by Anand et al (ref 171) but fail to mention that these researchers found no involvement of A2TR in the response of rat DRG neurons to mycolactone and instead attributed the effects to neurite degeneration. This supports the in vivo data from En et al (ref 66) and the discrepancies with the work of others should be discussed.
5. Therapeutics. Although to my knowledge there have been no studies with analogues there are a number of papers that have investigated the antiviral and anticancer activity of mycolactone. These should be mentioned and the potential for the use of less generally toxic analogues in this respect should be discussed. Regarding anti-cancer activity, it is very unclear whether Section 5.3.3 Mycolactone analogues as cytotoxic agents aims to draw attention to the potential usefulness of cytotoxicity of mycolactone derivatives in therapy or to highlight the negative aspects of these compounds as drugs. In either case, this paragraph is confusing and needs to be rewritten in a more explicit way.
Comments on the Quality of English Language
The English is adequate but there are numerous minor grammatical mistakes which sometimes make it hard to understand.
Author Response
Reviewer 2
This paper is a review of the chemistry and biological effects of mycolactone, a bacterial toxin produced by the causative agent of Buruli ulcer, Mycobacterium ulcerans. The paper covers the background on the disease, the chemistry and synthesis of mycolactone and its analogues, biological mechanism of action, the use of mycolactone in diagnosis and its potential as an analgesic and anti-inflammatory agent. While some aspects are clear and well written, overall, the review seems quite dated. The main focus is on the chemical synthesis of mycolactone and while this work was groundbreaking, the most recent paper cited is from 2017 and much of the ground has been covered in previous reviews (eg Gehringer and Altmann, Beilstein J Org Chem 2017, 13, 1596). In other areas, such as mechanism of action, diagnostics and therapeutics there are many omissions of relevant recent papers and there is a tendency to just summarize reports and ignore discrepancies between them when a critical analysis would be more valuable to the reader.
Minor concerns
- Figure 1: Image described as Plague should read Plaque.
The work Plague has been corrected to Plaque
- There is a lot of unnecessary repetition eg line 51 is almost identical to line 57. This is a long review and could be cut down considerably by removing such repeats.
The sentences have been modified by deleting portions to avoid the repetition noticed by the reviewer
- Line 78 and 89: should be Speke not Specke.
Corrections “Specke” to “Speke” on both lines have been corrected
- Mycolactone is sometimes referred to in the singular and sometimes in the plural eg paragraph starting on line 185 uses both interchangeably. This makes sense where different natural or synthetic analogues are used but this is not the case for many of the reports cited.
Careful consideration has been given to this comment and where appropriate, “mycolactone” was used in singular while “mycolactones” was used where different natural or synthetic analogues have been described.
- Line 330-31: No citation for Gurjar or Dai.
Citations for both Gurjar and Dai have been inserted.
- Line 614: reference 182 is wrong.
The wrong reference has been deleted and replaced with 65 and 66
- Line 726. The author’s state that 100% rounding at a concentration of 10µM at 48hr is comparable to a 90% rounding at 40nM at 24hr. The values are not comparable, and the two assays are totally different and should not be compared in this way.
- Line 769: binding potency to what?
Binding to WASP/N-WASP. This has been specified in the text
Major concerns
- Diagnostics: the section on the use of mycolactone for diagnosis discusses TLC and aptamer methods but does not cover the recent advances in antibody dependent detection using either ELISA or lateral flow assays (Warryn et al, 2020 https://doi.org/10.1371/journal.pntd.0008357, Warryn et al, 2021, https://doi.org/10.4049/jimmunol.2001232, Sakakibara et al, 2024 https//doi: 10.1016/j.jctube.2024.100469). These should be fully discussed in the review.
Recent advances in mycolactone detection have now been appropriately discussed and sectioned under 4.1Detection of mycolactone using mass spectrometry, 4.2 Immunological assays, 4.3 Diagnosis of Buruli ulcer with RNA aptamers, and 4.4 Fluorescent-thin layer chromatography (f-TLC). Emphasis is made on current literature as recommended
- Mechanism of action: The authors do not mention the work described in ref 84 proposes a fourth target of mycolactone, mTOR. This should also be discussed in this section.
Inhibition of mTOR by mycolactone has been addressed a subsection under Cellular targets responsible for the mechanism of action of mycolactone. Relevant literature has been cited.
- Mechanism of action: The authors give the impression that Sec61 inhibition is solely responsible for the immunosuppressive and anti-inflammatory actions of mycolactone. This is misleading as many of the changes in cell morphology and adhesion properties can be also attributed to this mechanism. Furthermore, the identification of Sec61 as a target using either chemical (ref 180) or CRISPR (Kawashima et al 2022 https://doi.org/10.1371/journal.pntd.0010672) based screens shows that Sec61 inhibition is a major factor in mycolactone driven cytotoxicity. To date, Sec61 is also the only target for which an interaction with mycolactone has been backed up with structural data (Gerard et al, 2020 https://doi: 10.1016/j.molcel.2020.06.013, Itskanov et al, 2023 doi: 10.1038/s41589-023-01337-y). These are backed up by molecular dynamics studies from the Swanson lab which demonstrate a preference for ER membranes and suggest the differences between mycolactone A and B interaction with membranes and the translocon could explain their different potencies (Nguyen et al, 2023 https://doi: 10.3390/toxins15080486 and da Hora et al, 2022 htpps://doi: 10.1016/j.bpj.2022.10.019). These are highly relevant studies that should be included in the review.
All relevant literature on sec61 mechanism of action as a major cellular target are addressed. We reviewed the use of electron cryo-microscopy by Gérard et al. to analyze translocon and identify the structure of a mycolactone-inhibited mammalian Sec61. We review the free-energy investigations and molecular dynamics simulations to characterize the association and penetration of mycolactone via the plasma membranes.
- Analgesic effects of mycolactone. The authors cite a paper by Anand et al (ref 171) but fail to mention that these researchers found no involvement of A2TR in the response of rat DRG neurons to mycolactone and instead attributed the effects to neurite degeneration. This supports the in vivo data from En et al (ref 66) and the discrepancies with the work of others should be discussed.
We discussed mycolactone-induced analgesic effects in rat and human primary dorsal root ganglion (DRG) sensory neurons and attributed the mechanism of action to neurite degeneration and not A2TR.
- Therapeutics. Although to my knowledge there have been no studies with analogues there are a number of papers that have investigated the antiviral and anticancer activity of mycolactone. These should be mentioned and the potential for the use of less generally toxic analogues in this respect should be discussed. Regarding anti-cancer activity, it is very unclear whether Section 5.3.3 Mycolactone analogues as cytotoxic agents aims to draw attention to the potential usefulness of cytotoxicity of mycolactone derivatives in therapy or to highlight the negative aspects of these compounds as drugs. In either case, this paragraph is confusing and needs to be rewritten in a more explicit way.
Aside discussing the potential of mycolactone and its analogues as analgesic and immunosuppressants, we these comments by discussing other therapeutic applications of mycolactone and its derivatives by considering their anti-cancer and anti-viral potentials. It is sectioned 5.3.3 Other therapeutic applications of mycolactone derivatives.
Reviewer 3 Report
Comments and Suggestions for Authors
The authors have written a very comprehensive review of mycolactone that covers historical aspects of its discovery and the disease it causes, the biosynthetic pathway and chemical synthetic routes for its production. A lot of focus is given to the Kishi group’s work, but other who have contributed have been appropriately cited too. They also discuss its biological effects and potential applications in medicine (including of mycolactone analogues.
A general point is a lot of the figures have been reproduced from literature sources and so all of the necessary permissions will need to be sought for their republication.
Another general point – there should be no space between numbers and % - it is not a unit – it is part of the number.
Figure 2: the text in the figure caption within the original figure is too small – reproduce the figure and add your own caption with acknowledgement of the origin of the figure.
line53: “where the reverse is true” – it is not clear what is meant here – please clarify.
line 182: please change to “a 12-membered ring polyketide”
line 209: the “nothern acyl side-chain” needs to be indicated explicitly in Figure 3 – don’t assume the reader will understand what is meant by this term.
Line 266: Figure 5a is referred to immediately after the phrase “these genes possess very high DNA sequence identity”, but figure 5a does not show anything relating to DNA sequence similarity. It might be the authors meant figure 3a, but it also does not show DNA sequence similarity
Line 281: similarly, the placing of the reference to Figure 5b (presumably meant to be Figure 3b) is not appropriate as it does not show oxidation of the side chain. The authors need to pick their figures more carefully to support the points they are making in the text.
Figure 3 – the text is too small to read – if this figure is really necessary it needs to be redrawn with larger text.
Scheme 1, 2 and 3 – the reagents are listed in the caption unlike the later schees where they are listed in the figure. I think the latter approach is preferable, but the authors also need to show at least one more intermediate structure in each of schemes 1-3 as listing reagents for 5 or 6 steps is too big a conceptual leap for the reader to appreciate what is happening here.
line 414: the section title says the work was from 2012, but reference 125 was published in 2010.
lines 519-526: The points here are a bit repetitive – the text would benefit from being simplified.
Line 549: The Aptamers described will have been selected from a library and so the correct way to describe this is “Aptamers have been selected that posses high affinity...”
Line 569: “underlining” might be better to be “underlying”
Line 574: here is says that mycolactone inhibits WASP, but in later sections it is described as being a WASP activator
Figure 10 needs to be redrawn – the structures are too small to be seen.
Line 583-585: there is no need to keep repeating definitions of WASP and N-WASP as this was just done in the previous paragraph
Line 616 - the text is quite repetitive on sec61-dependent transloctaion – the point has just been made earlier in the paragraph.
line 646-649 – i twould be better to acknowledge that you have already discussed this earlier in the review.
line 692 – remove “totally”
line 787-788: Aspirin is an NSAID
Comments on the Quality of English Language
The quality of English is good, but as highlighted in my review there are places where the text becomes quite repetitive and I suggest another close reading would allow it to be made more concise.
Author Response
Reviewer 3
The authors have written a very comprehensive review of mycolactone that covers historical aspects of its discovery and the disease it causes, the biosynthetic pathway and chemical synthetic routes for its production. A lot of focus is given to the Kishi group’s work, but other who have contributed have been appropriately cited too. They also discuss its biological effects and potential applications in medicine (including of mycolactone analogues.
A general point is a lot of the figures have been reproduced from literature sources and so all of the necessary permissions will need to be sought for their republication.
All figures have been appropriately sourced and cited. However, steps will be taken to seek for permission to reproduce figures.
Another general point – there should be no space between numbers and % - it is not a unit – it is part of the number.
This has been addressed across the entire manuscript.
Figure 2: the text in the figure caption within the original figure is too small – reproduce the figure and add your own caption with acknowledgement of the origin of the figure.
An enhanced figure has been sourced from World Health Organization with the small text in the figure removed. All information is captured in the caption.
line53: “where the reverse is true” – it is not clear what is meant here – please clarify.
The sentence has been clarified and changed to …. where the disease is more predominant in the adult population
line 182: please change to “a 12-membered ring polyketide”
This change has been made.
line 209: the “nothern acyl side-chain” needs to be indicated explicitly in Figure 3 – don’t assume the reader will understand what is meant by this term.
The northern acyl side chain has been indicated on Figure 3 as recommended by the reviewer.
Line 266: Figure 5a is referred to immediately after the phrase “these genes possess very high DNA sequence identity”, but figure 5a does not show anything relating to DNA sequence similarity. It might be the authors meant figure 3a, but it also does not show DNA sequence similarity
The correct labelling should have been Figure 3 and this has been corrected. The phrase “these genes possess very high DNA sequence identity” has also been deleted to avoid any ambiguity.
Line 281: similarly, the placing of the reference to Figure 5b (presumably meant to be Figure 3b) is not appropriate as it does not show oxidation of the side chain. The authors need to pick their figures more carefully to support the points they are making in the text.
The correct labelling should have been Figure 3 and this has been corrected.
Figure 3 – the text is too small to read – if this figure is really necessary it needs to be redrawn with larger text.
The picture has been enhanced and the font size of the text has been increased.
Scheme 1, 2 and 3 – the reagents are listed in the caption unlike the later schemes where they are listed in the figure. I think the latter approach is preferable, but the authors also need to show at least one more intermediate structure in each of schemes 1-3 as listing reagents for 5 or 6 steps is too big a conceptual leap for the reader to appreciate what is happening here.
The reagents have been listed in the caption for all the scheme in the manuscript
line 414: the section title says the work was from 2012, but reference 125 was published in 2010.
The years have been removed to avoid the confusion in when the specific synthesis was conducted.
lines 519-526: The points here are a bit repetitive – the text would benefit from being simplified.
Sentence modified to avoid repetitions
Line 549: The Aptamers described will have been selected from a library and so the correct way to describe this is “Aptamers have been selected that posses high affinity...”
The sentence has been modified to include “Aptamers have been selected that possess high affinity...
Line 569: “underlining” might be better to be “underlying”
The work “underlining" has been changed to “underlying”
Line 574: here is says that mycolactone inhibits WASP, but in later sections it is described as being a WASP activator
Both terms have been used in describing the MoA of Aptamers. In reference 178 they are said to “block/inhibit the activity of Sec61” while in reference 182, they are said to “activate WASP”
Figure 10 needs to be redrawn – the structures are too small to be seen.
Figure 10 has been improved and more visible
Line 583-585: there is no need to keep repeating definitions of WASP and N-WASP as this was just done in the previous paragraph
This has been corrected
Line 616 - the text is quite repetitive on sec61-dependent transloctaion – the point has just been made earlier in the paragraph.
This section was describing the “. Cellular targets responsible for the mechanism of action of mycolactone” hence the repetition.
line 646-649 – It would be better to acknowledge that you have already discussed this earlier in the review.
We have now acknowledged that this was discussed earlier
line 692 – remove “totally”
“Totally” was removed as recommended
line 787-788: Aspirin is an NSAID
NSAID was inserted after Aspirin
Reviewer 4 Report
Comments and Suggestions for Authors
This review provides a comprehensive overview of mycolactone, its biosynthesis, and the potential therapeutic and diagnostic applications relevant to Buruli ulcer research. The manuscript effectively summarizes a vast array of studies on mycolactone’s chemical synthesis, biological pathways, and mechanisms of action. The organization of topics from biosynthetic origins to applied medical insights creates a structured narrative that is beneficial for readers.
The review excels in presenting the dual therapeutic and diagnostic potential of mycolactone, which is a rare combination that fills a notable gap in the current literature. Additionally, the manuscript’s inclusion of non-target and safety considerations when assessing therapeutic uses enhances its relevance for researchers aiming to develop safe, effective treatments.
Major Comments
Figures and Illustrations:
Clarity and Structure: Figures need clearer labels and consistent formatting. Ensure all figures are directly relevant, well-labeled, and clearly referenced in the text. Some figures could benefit from enhanced captions that offer sufficient standalone information for a reader unfamiliar with the content.
Quality and Resolution: Check that all images and figures meet the journal’s quality and resolution standards, ensuring there’s no pixelation or blurring in the printed versions.
Consistency: Ensure that all symbols, notations, and abbreviations are consistent throughout the figures and corresponding text. This alignment helps in improving readability and coherence across sections.
English Grammar and Flow:
Sentence Structure: Several sentences are overly complex, leading to readability issues. Consider breaking them down into shorter, more direct sentences to improve clarity.
Novelty and Scope:
Emphasis on Novel Contributions: Although the manuscript comprehensively reviews existing knowledge, it lacks a clear emphasis on what new insights this review offers to the field. Add sections or statements that clarify how this review advances the understanding of mycolactone’s potential therapeutic applications or innovative diagnostic methodologies.
The flow between sections could be improved by creating smoother transitions, especially when moving from one synthesis pathway to another or comparing diagnostic methods. Use transition sentences that guide the reader through the various sections logically.
Conclusion and Future Directions: Strengthen the conclusion by summarizing the main insights and proposing concrete future directions or applications of the findings. Consider adding a roadmap for future research areas where knowledge gaps exist, especially in therapeutic applications.
Rephrase High-Similarity Sections: Paraphrase areas that closely resemble prior publications, especially in the introduction, literature review, and methodology sections. This could involve rewording sentences to reflect the manuscript's perspective and avoid direct overlap.
Comments on the Quality of English LanguageModerate editing of English language required.
Author Response
Reviewer 4
This review provides a comprehensive overview of mycolactone, its biosynthesis, and the potential therapeutic and diagnostic applications relevant to Buruli ulcer research. The manuscript effectively summarizes a vast array of studies on mycolactone’s chemical synthesis, biological pathways, and mechanisms of action. The organization of topics from biosynthetic origins to applied medical insights creates a structured narrative that is beneficial for readers.
The review excels in presenting the dual therapeutic and diagnostic potential of mycolactone, which is a rare combination that fills a notable gap in the current literature. Additionally, the manuscript’s inclusion of non-target and safety considerations when assessing therapeutic uses enhances its relevance for researchers aiming to develop safe, effective treatments.
Major Comments
Figures and Illustrations:
Clarity and Structure: Figures need clearer labels and consistent formatting. Ensure all figures are directly relevant, well-labeled, and clearly referenced in the text. Some figures could benefit from enhanced captions that offer sufficient standalone information for a reader unfamiliar with the content.
This has been addressed across the entire manuscript.
Quality and Resolution: Check that all images and figures meet the journal’s quality and resolution standards, ensuring there’s no pixelation or blurring in the printed versions.
Consistency: Ensure that all symbols, notations, and abbreviations are consistent throughout the figures and corresponding text. This alignment helps in improving readability and coherence across sections.
The quality and resolution of images have been enhanced
English Grammar and Flow:
Sentence Structure: Several sentences are overly complex, leading to readability issues. Consider breaking them down into shorter, more direct sentences to improve clarity.
The sentences that are too long have been split or rephrased for clarity
Novelty and Scope:
Emphasis on Novel Contributions: Although the manuscript comprehensively reviews existing knowledge, it lacks a clear emphasis on what new insights this review offers to the field. Add sections or statements that clarify how this review advances the understanding of mycolactone’s potential therapeutic applications or innovative diagnostic methodologies.
In this manuscript, we highlight from the biosynthetic origin of mycolactone, its chemical synthesis and medical insights for the benefit of readers by summarizing the chemical synthesis, biological pathways, and mechanisms of action.
The flow between sections could be improved by creating smoother transitions, especially when moving from one synthesis pathway to another or comparing diagnostic methods. Use transition sentences that guide the reader through the various sections logically.
An attempt was made to address this comment by the reviewer
Conclusion and Future Directions: Strengthen the conclusion by summarizing the main insights and proposing concrete future directions or applications of the findings. Consider adding a roadmap for future research areas where knowledge gaps exist, especially in therapeutic applications.
This comment has been addressed
Rephrase High-Similarity Sections: Paraphrase areas that closely resemble prior publications, especially in the introduction, literature review, and methodology sections. This could involve rewording sentences to reflect the manuscript's perspective and avoid direct overlap.
This comment and recommendations have been addressed.
Reviewer 5 Report
Comments and Suggestions for Authors
1. If figures are not original, copyrights will be needed.
2. Figure labels are not in order. On page 7, there’s repeated figure 1, 2, 3...
3. Numbers in chemical formulas should be subscript, please check the whole context including figures.
4. Formatting of schemes are not consistent, eg: font format
5. Figure 10 is in low-resolution, please provide an improved one.
Author Response
Reviewer 5
- If figures are not original, copyrights will be needed.
All figures have been appropriately sourced and cited. However, steps will be taken to seek for permission to reproduce figures.
- Figure labels are not in order. On page 7, there’s repeated figure 1, 2, 3...
The figure labels have been modified to continue from 1-14
- Numbers in chemical formulas should be subscript, please check the whole context including figures.
Numbers of chemical formulas have all been changed to subscripts
- Formatting of schemes are not consistent, eg: font format
All the schemes have been formatted consistently
- Figure 10 is in low-resolution, please provide an improved one.
The resolution of Figure 10 has been improved.
Round 2
Reviewer 2 Report
Comments and Suggestions for Authors
The paper is much improved. The concerns raised in the initial review have been addressed.
Comments on the Quality of English LanguageThe English is adequate but there is still a degree of repetition and many minor grammatical mistakes. I suggest the authors go through the paper carefully to check for these and to make sure that they are saying what they intend to say.
Author Response

(The authors gave the same response as above.)
